# A Review on the Role of miR-149-5p in the Carcinogenesis

**DOI:** 10.3390/ijms23010415

**Published:** 2021-12-30

**Authors:** Soudeh Ghafouri-Fard, Tayyebeh Khoshbakht, Bashdar Mahmud Hussen, Sepideh Kadkhoda, Mohammad Taheri, Arash Tafrishinejad

**Affiliations:** 1Department of Medical Genetics, School of Medicine, Shahid Beheshti University of Medical Sciences, Tehran 16666-63111, Iran; s.ghafourifard@sbmu.ac.ir; 2Men’s Health and Reproductive Health Research Center, Shahid Beheshti University of Medical Sciences, Tehran 16666-63111, Iran; sare.khoshbakht@gmail.com; 3Department of Pharmacognosy, College of Pharmacy, Hawler Medical University, Kurdistan Region, Erbil 44001, Iraq; Bashdar.Hussen@hmu.edu.krd; 4Department of Medical Genetics, School of Medicine, Tehran University of Medical Sciences, Tehran 16666-63111, Iran; kadkhoda.sepideh@yahoo.com; 5Institute of Human Genetics, Jena University Hospital, 07740 Jena, Germany; 6Skull Base Research Center, Loghman Hakim Hospital, Shahid Beheshti University of Medical Sciences, Tehran 16666-63111, Iran

**Keywords:** miR-149-5p, cancer, biomarker, expression, ncRNAs

## Abstract

miR-149 is an miRNA with essential roles in carcinogenesis. This miRNA is encoded by the *MIR149* gene on 2q37.3. The miR-149 hairpin produces miR-149-5p and miR-149-3p, which are the “guide” and the sister “passenger” strands, respectively. Deep sequencing experiments have shown higher prevalence of miR-149-5p compared with miR-149-3p. Notably, both oncogenic and tumor suppressive roles have been reported for miR-149-5p. In this review, we summarize the impact of miR-149-5p in the tumorigenesis and elaborate mechanisms of its involvement in this process in a variety of neoplastic conditions based on three lines of evidence, i.e., in vitro, in vivo and clinical settings.

## 1. Introduction

MicroRNAs (miRNAs) are a group of small-sized regulatory noncoding RNAs which bind to the 3′-UTR of target mRNAs in a specific manner to inhibit their translation [1]. miRNAs can affect diverse aspects of carcinogenesis, tumor evolution, metastatic and angiogenic processes, and resistance to chemoradiotherapeutics [2]. 

miR-149 is an miRNA with essential roles in carcinogenesis. This miRNA is encoded by *MIR149* gene on 2q37.3. The miR-149 hairpin produces miR-149-5p and miR-149-3p which are the “guide” and the sister “passenger” strands, respectively [3]. These two miRNAs have completely dissimilar sequences, suggesting their distinct roles in biological processes [3]. Deep sequencing experiments have shown a higher prevalence of miR-149-5p compared with miR-149-3p.

Physiological functions of miR-149-5p have been evaluated by researchers. For instance, it has been shown to be up-regulated in bovine adipocytes in certain times during their differentiation. The functional role of miR-149-5p in this process is exerted through targeting CRTC1 and CRTC2 [4]. Moreover, the expression of miR-149-5p has been found to be decreased in PDGF-BB-induced vascular smooth muscle cells. Up-regulation of miR-149-5p could suppress the proliferation, invasion and migration of vascular smooth muscle cells, whereas its silencing has the opposite effect. Moreover, histone deacetylase 4 (HDAC4) is the miR-149-5p target, through which this function of miR-149-5p is accomplished [5].

Notably, both oncogenic and tumor suppressive roles have been reported for miR-149-5p. In this review, we summarize the impact of miR-149-5p in tumorigenesis and elaborate mechanisms of its involvement in this process in a variety of neoplastic conditions based on three lines of evidence, i.e., in vitro, in vivo and clinical settings.

## 2. Cell Line Studies

### 2.1. Gastrointestinal Tumors

A high throughput sequencing experiment in gastric cancer has shown down-regulation of the circular RNA (circRNA) circNRIP1. This circRNA has been shown to act as a sponge for miR-149-5p. This miRNA regulates metabolic pathways through AKT1/mTOR axis. miR-149-5p suppression has been shown to exert similar effects with up-regulation of circNRIP1 in gastric cancer cells [6]. Another study in gastric cancer has shown down-regulation of miR-149-5p parallel with the up-regulation of the long non-coding RNA (lncRNA) BLACAT1 and KIF2A. miR-149-5p has been shown to be sponged by BLACAT1, leading to the up-regulation of KIF2A [7]. Moreover, miR-149-5p has been demonstrated to be sponged by circNHSL1. In fact, the effects of circNHSL1 on cancer cell migration, invasiveness and glutaminolysis is mediated through sponging miR-149-5p. Experiments in gastric cancer cells have led to identification of YWHAZ as the target of miR-149-5p [8].

Two independent studies have shown the sponging effects of LINC00460 on miR-149-5p in the context of colorectal cancer. LINC00460/miR-149-5p has important functions in the regulation of expression of p53 [9] and BGN [10] in this type of cancer.

In addition, miR-149-5p has been found to be sponged by circCTNNA1, a circRNA with pro-proliferative and pro-migratory effects in colorectal cancer cells. miR-149-5p has been shown to decrease the expression of FOXM1. Thus, the circCTNNA1/miR-149-5p/FOXM1 axis has been suggested as a target for therapeutic interventions in colorectal cancer [11]. miR-149-5p has also been shown to be sponged by other oncogenic non-coding RNAs such as PCAT1 [12], DLGAP1-AS1 [13] and circ5615 [14] in colorectal cancer cells.

### 2.2. Thyroid Cancer

miR-149-5p has a tumor suppressor role in medullary thyroid carcinoma through directly targeting GIT1. miR-149-5p up-regulation could inhibit the proliferation and invasive properties of medullary thyroid carcinoma cells [15]. In papillary thyroid cancer, miR-149-5p has been shown to bind with circ-FLNA, a circRNA whose expression is regulated by TR4. Binding of circ-FLNA with this miRNA releases MMP9 from the inhibitory effects of miR-149-5p. Thus, the TR4/circ-FLNA/miR-149-5p/MMP9 axis has been identified as an important player in the pathogenesis of papillary thyroid cancer [16].

### 2.3. Oral Cancer

Expression of miR-149-5p has been shown to be reduced in oral squamous cell carcinoma cells, particularly in cisplatin resistant cells. Up-regulation of miR-149-5p could enhance the cytotoxic effects of this drug in both resistant cells and parental cells. miR-149-5p could also decrease the proliferation, migratory aptitude and invasiveness of both cell lines, and promote their apoptosis through decreasing expression of TGFβ2 [17]. In this kind of cancer, miR-149-5p has been shown to be sponged by DLEU1 lncRNA. Since miR-149-5p could decrease the expression of CDK6, down-regulation of miR-149-5p by this lncRNA leads to up-regulation of CDK6 and a subsequent increase in cell proliferation and cell cycle progression [18].

### 2.4. Ovarian Cancer

In ovarian cancer, two different studies have shown contradictory results about the role of miR-149-5p. Xu et al. have shown that miR-149-5p silencing increases the sensitivity of ovarian cancer cells to cisplatin. miR-149-5p has been found to target the most important kinase elements of the Hippo signaling pathway, i.e., MST1 and SAV1, leading to the inactivation of TEAD expression [19]. Conversely, Li et al. have shown a tumor suppressor role for miR-149-5p in ovarian cancer cells through targeting FOXM1 [20].

### 2.5. Osteosarcoma

Expression of miR-149-5p has been found to be decreased in osteosarcoma cells. Forced up-regulation of miR-149-5p has inhibited the growth of these cells through targeting TNFRSF12A. The anti-proliferative impacts of miR-149-5p are exerted through modulation of the PI3K/AKT pathway [21].

### 2.6. Breast Cancer

Expression of miR-149-5p has been found to be down-regulated in breast cancer cells, parallel with the up-regulation of circ_0072995 and SHMT2 levels. This miRNA has a role in the regulation of anaerobic glycolysis through the suppression of SHMT2 expression [22].

Another experiment in breast cancer cells has shown that the anesthetic agent propofol can alter resistance to trastuzumab via modulation of the IL-6/miR-149-5p molecular axis. Authors have reported production of high levels of IL-6 and IL-8 cytokines, were released by resistant cells, induction of the stemness phenotype mammospheres and enhancement of epithelial-mesenchymal transition (EMT) in trastuzumab resistant cells. Notably, propofol could inhibit all of these processes through the up-regulation of miR-149-5p and subsequent down-regulation of IL-6 expression [23]. Another study has shown the effects of ursolic acid in the attenuation of the paclitaxel resistance phenotype in breast cancer cells through influencing the miR-149-5p/myd88 axis [24].

### 2.7. Urogenital Cancers

In prostate cancer cells, hsa-miR-149-5p has been found to suppress tumorigenic processes through inhibiting expression of RGS17 [25].

In bladder cancer cells, the anti-cancer effect of miR-149-5p is exerted through the inhibition of expression of RNF2. Notably, miR-149-5p has been shown to be sponged by the oncogenic circRNA_100146 in these cells [26]. Figure 1 shows the tumor-suppressive role of miR-149-5p in esophageal cancer, bladder cancer, breast cancer, ovarian cancer, thyroid cancer and colorectal cancer.

### 2.8. Lung Cancer

In lung cancer cells, miR-149-5p has been found to down-regulate the expression of B3GNT3, an oncogene that influences lung cancer cell proliferation and invasiveness [27]. Moreover, expression of this miRNA has been found to be decreased by HOTAIR [28], HNF1A-AS1 [29] and MIAT [30] lncRNAs in these cells. In fact, HOTAIR has been shown to induce cisplatin resistance and increase the proliferation, migratory potential and invasiveness of cisplatin-resistant lung cancer cells through targeting miR-149-5p [31].

Another study on lung cancer cells has shown the role of miR-149-5p in the enhancement of the response to gefitinib. The oncogenic lncRNA LINC00460 has been found to promote resistance to EGFR-TKI through sequestering miR-149-5p, thus increasing IL-6 levels and facilitating EMT process [32]. Conversely, comparison of miRNA profiles in gefitinib-resistant human lung cancer cells and the parental cells has shown up-regulation of miR-149-5p in resistant cells. miR-149-5p mimics could reduce the motility of lung cancer cells. Up-regulation of miR-149-5p could efficiently evaluate the half maximal inhibitory concentrations of the cell following treatment with gefitinib. Besides, expressions of miR-149-5p in both cell lines have been inversely correlated with caspase-3 levels. Taken together, miR-149-5p expression is increased in the gefitinib-resistant human lung cancer cells contributing in the acquired resistance to gefitinib [33].

Moreover, over-expression of miR-149-5p in cancer-derived exosomes could enhance growth of tumor cells and inhibit their apoptosis through suppression of AMOTL2 levels [34].

### 2.9. Hepatocellular Carcinoma

In hepatocellular carcinoma, M2 macrophages have been shown to increase the invasiveness and migratory potential of cancer cells through decreasing miR-149-5p levels and subsequent activation of MMP9 signaling [35]. In this kind of cancer, miR-149-5p can enhance response to sorafenib through decreasing AKT1 levels [36]. Moreover, it can exert anti-cancer effects through decreasing MAP2K1 expression [37].

miR-149-5p has also been shown to target 3’-UTR of the *MTHFR* transcript. Expression of miR-149-5p has been shown to be increased in both normal hepatocytes and hepatocellular carcinoma cells in response to folic acid deficiency. However, this condition has resulted in different responses in the expression of MTHFR in these two cell lines. In fact, MTHFR levels are reduced in cancerous cells but remained constant in normal hepatocytes in response to folic acid deficiency. Thus, miR-149-5p exerts distinct post-transcriptional influences on this gene upon folic acid deficiency in normal hepatocytes and hepatocellular carcinoma cells. Moreover, this miRNA may have an anti-cancer effect in longstanding folic acid deficiency conditions [38].

In esophageal squamous cell carcinoma, miR-149-5p exerts anti-cancer effects through the modulation of IL-6/STAT3 axis [39]. Figure 2 shows the tumor-suppressive role of miR-149-5p in gastric cancer, lung cancer, colorectal cancer, hepatocellular carcinoma and oral squamous cell carcinoma.

Table 1 summarizes the results of studies which assessed the role of miR-149-5p or miR-149-5p-interacting genes in carcinogenesis in cell lines.

## 3. Animal Studies

In gastric cancer, in vivo studies have indirectly confirmed the tumor suppressor role of miR-149-5p. In fact, the silencing of circNRIP1 [6], BLACAT1 [7] or circNHSL1 [8] which sequester this miRNA has led to a reduction of tumor size in animal models. Similarly, silencing of LINC00460 [9], circCTNNA1 [11], DLGAP1-AS1 [13] or circ5615 [14] has attenuated tumor growth in animal models of colorectal cancer.

In vivo studies have shown the impact of miR-149-5p in the facilitation of the inhibitory function of propofol against lung metastases in a breast cancer xenograft model [23]. On the other hand, miR-149-5p has been shown to exert oncogenic effects in ovarian cancer, since its silencing has decreased tumor volume and resistance to cisplatin [19]. Table 2 shows the role of miR-149-5p or miR-149-5p-intercating genes in carcinogenesis based on animal models.

## 4. Clinical Studies

Independent studies in gastric, liver, colorectal, medullary/papillary thyroid, breast, prostate, esophageal, renal, cervical and oral squamous cell cancers as well as osteosarcoma have shown down-regulation of miR-149-5p or its sequestering lncRNAs/circRNAs, thus providing evidence for tumor suppressor role of miR-149-5p (Table 3). In gastric cancer, up-regulation of circNRIP1 (which sponges miR-149-5p) has been associated with shorter overall survival and disease-free survival (DFS) times [6]. In colorectal cancer, up-regulation of LINC00460 (which sponges miR-149-5p) has been associated with poorer DFS [10].

In bladder cancer, both tumor suppressor [26] and oncogenic [49] effects have been reported for mR-149-5p. In ovarian cancer, miR-149-5p has been shown to be up-regulated in cancerous tissues, particularly chemoresistant ones compared with controls [19]. However, another study has shown up-regulation of circPVT1, a miR-149-5p-sequestering circRNA in this type of cancer [20]. In lung cancer, while most studies have indicated a tumor suppressor role for miR-149-5p (Table 3), assessment of a GEO dataset has shown up-regulation of this miRNA in cancer patients compared with controls [34].

**Table 3 ijms-23-00415-t003:** Role of miR-149-5p or miR-149-5p-interacting genes in carcinogenesis based on clinical studies (OS: Overall survival, DFS: disease-free survival, PFS: progression-free rate TNM: tumor-node-metastasis, ANCTs: adjacent non-cancerous tissues, NSCLC: non-small cell lung cancer, ccRCC: clear cell renal cell carcinoma).

Tumor Type	Samples	Expression of miR-149-5p or Other Genes (Tumor vs. Normal)	Kaplan-Meier Analysis (Impact of miR-149-5p Dysregulation or Other Genes Dysregulation)	Association of Expression of miR-149-5p or Expression of Other Genes with Clinicopathologic Characteristics	Reference
Gastric cancer (GC)	3 pairs of GC tissues and ANCTs	Up-regulation of circNRIP1 (which sponges miR-149-5p)	shorter OS and DFS	GC tumor size and lymphatic invasion	[6]
	40 GC patients and 40 normal controls	Up-regulation of circNRIP1 (which sponges miR-149-5p)	_	_	
	52 pairs of GC tissues and ANCTs	Up-regulation of BLACAT1 (which sponges miR-149-5p)	_	WHO grade and TNM stage	[7]
	20 GC patients and 20 normal controls	Up-regulation of CircNHSL1 (which sponges miR-149-5p)	_	tumor size, TNM stages, lymphatic metastasis, and distant metastasis	[8]
	GEO databases: (GSE23739, GSE26595, GSE26645, GSE28700, GSE33743, GSE54397, GSE63121, GSE78091, and GSE93415	Down-regulation of miR-149-5p	_	distal stomach	[50]
	TCGA database	Down-regulation of miR-149-5p	_	_	
Stomach Adenocarcinoma (STAD)	serum samples from 130 STAD patients and 116 healthy controls	Down-regulation of miR-149-5p	_	_	[51]
	serum samples from 30 STAD patients and 24 healthy controls	Down-regulation of miR-149-5p	_	_	
Colorectal cancer (CRC)	21 pairs of CRC tissues and ANCTs	Up-regulation of LINC00460 (which sponges miR-149-5p)	_	clinical stage and node status	[9]
	TCGA database	Up-regulation of LINC00460 (which sponges miR-149-5p)	_	_	
	40 pairs of CRC tissues and ANCTs	Up-regulation of LINC00460 (which sponges miR-149-5p)	Poorer DFS	_	[10]
	60 pairs of colon cancer tissues and ANCTs	Up-regulation of circCTNNA1 (which sponges miR-149-5p), Up-regulation of FOXM1	_	advanced TNM stage	[11]
	180 pairs of CRC tissues and ANCTs	Up-regulation of circCTNNA1 (which sponges miR-149-5p)	lower OS	_	
	55 pairs of CRC tissues and ANCTs	Down-regulation of miR-149-5p	_	_	[12]
	42 pairs of CRC tumor tissues and ANCTs	Up-regulation of DLGAP1-AS1 (which sponges miR-149-5p)	lower OS	advanced clinical stage (phase III–IV)	[13]
	GSE142837 analysis	Up-regulation of circ5615 (which sponges miR-149-5p)	_	_	[14]
	99 pairs of CRC tissues and ANCTs	Up-regulation of circ5615 (which sponges miR-149-5p)	shorter OS	higher T stage	
	TCGA and GEO dataset: GSE33113 and GSE41328	Up-regulation of LINC00460 (which sponges miR-149-5p)	shorter OS	larger tumor sizes, advanced TNM stages, and lymph node metastasis	[40]
Medullary thyroid carcinoma (MTC)	36 42 pairs of MTC tumor tissues and ANCTs	Down-regulation of miR-149-5p	shorter OS	distant metastases and TNM stage	[15]
Papillary thyroid cancer (PTC)	20 pairs of primary PTC tumors and the paired lymph node metastatic PTC tumors	Up-regulation of TR4	shorter OS	nodal metastasis	[16]
Oral squamous cell carcinoma (OSCC)	34 pairs of OSCC tissues and ANCTs	Down-regulation of miR-149-5p	_	_	[17]
	10 pairs of OSCC tissues and ANCTs	Up-regulation of DLEU1 (which sponges miR-149-5p)	_	advanced stage	[18]
Ovarian cancer	TCGA datasets	Up-regulation of miR-149-5p in chemoresistant tissues	_	_	[19]
	20 ovarian cancer tissues and 10 benign ovarian lesion tissues	Up-regulation of miR-149-5p in ovarian cancer tissues than benign ovarian tissues	_	_	
	GTEx database	Up-regulation of circPVT1	shorter PFS	stage 3/4 and grade 3 OV	[20]
Osteosarcoma	191 sarcoma patients and 66 adjacent normal samples	Down-regulation of miR-149-5p	shorter OS	age and tumor size	[21]
Breast cancer	70 pairs of breast cancer tissues and ANCTs	Up-regulation of circ_0072995	_	_	[22]
	1104 breast cancer tissues and 113 adjacent normal tissues	Up-regulation of SHMT2	_	_	
Lung cancer	TCGA dataset	Up-regulation of B3GNT3 (a target of miR-149-5p)	lower OS and DFS	advanced TNM stages	[27]
	120 pairs of lung cancer tissues and ANCTs	Up-regulation of B3GNT3 (a target of miR-149-5p)	lower OS and DFS	late TNM stages, bigger tumor size, distant metastasis and recurrence	
	GEO dataset: GSE19188 (91 NSCLC tissues and 65 ANCTs	Up-regulation of HOTAIR (which sponges miR-149-5p)	unfavorable DFS	_	[28]
	60 pairs of NSCLC tissues and ANCTs	Up-regulation of HNF1A-AS1 (which sponges miR-149-5p)	shorter OS	advanced TMN stage, big tumor size, and lymph node metastasis	[29]
	80 pairs of NSCLC and ANCTs	Up-regulation of MIAT (which sponges miR-149-5p)	shorter OS	advanced pathological stage	[30]
	35 DDP-resistant NSCLC tumors and 35 DDP-sensitive NSCLC tumors	Up-regulation of HOTAIR in DDP-resistant NSCLC tumors than DDP-sensitive NSCLC tumors	_	TNM stage, lymph node metastasis and DDP response	[31]
	GEO dataset: GSE111803 analysis (5 lung adenocarcinoma patients and 5 healthy controls)	Up-regulation of miR-149-5p	_	advanced clinical stage	[34]
	TCGA dataset	Up-regulation of LINC00460 in lung adenocarcinoma tissues with EGFR-activating mutations than in normal tissues	_	_	[32]
	62 patients with EGFR-mutant lung adenocarcinoma treated with EGFR-TKI	Up-regulation of LINC00460	shorter OS and PFS	_	
Hepatocellular carcinoma (HCC)	79 pairs of HCC tissues and ANCTs	Up-regulation of NEAT1 (which sponges miR-149-5p) Down-regulation of miR-149-5p	shorter OS and DFS	tumor stage, lymphatic metastasis, and sorafenib resistance	[36]
	48 pairs of HCC tissues and ANCTs	Up-regulation of PART1 (which sponges miR-149-5p)	_	_	[37]
	23 pairs of HCC tissues and ANCTs	Up-regulation of SNHG8 (which sponges miR-149-5p)	higher recurrence rates	_	[41]
Prostate carcinoma (PCa)	GEO DataSets: GSE17317 and GSE34932	Down-regulation of miR-149-5p	_	_	[25]
	30 pairs of PCa tissues and ANCTs	Down-regulation of miR-149-5p	_	_	
Esophageal squamous cell cancer (ESCC)	55 pairs of ESCC tissues and ANCTs	Up-regulation of circ_0000654 (which sponges miR-149-5p)	_	higher T stage and local lymph node metastasis	[39]
Bladder cancer	68 pairs of Bladder cancer tissues and ANCTs	Up-regulation of circRNA_100146 (which sponges miR-149-5p) Down-regulation of miR-149-5p	_	tumor stage, LN status, histological grade, and tumor size	[26]
Bladder cancer	10 Bladder cancer patients and 10 healthy controls	Up-regulation of miR-149-5p	_	_	[49]
	TCGA database	Up-regulation of miR-149-5p	poor OS	_	
	urine of 70 Bladder cancer patients and 90 healthy controls	Up-regulation of miR-149-5p	_	_	
Renal cell carcinoma (RCC)	32 pairs of RCC tissues and ANCTs	Down-regulation of miR-149-5p	_	_	[42]
	TCGA database (237 pairs of ccRCC tissues and ANCTs)	Down-regulation of miR-149-5p	shorter OS	_	[52]
	16 pairs of ccRCC tissues and ANCTs	Down-regulation of miR-149-5p	_	_	[43]
Cervical cancer (CC)	37 pairs of CC tissues and ANCTs	Up-regulation of circ_0075341 (which sponges miR-149-5p)	_	larger tumor size, advanced FIGO stage, and lymph-node metastasis	[44]
	GEO database: GSE102686	Up-regulation of circ_0011385 (which sponges miR-149-5p)	_	_	[45]
	50 pairs of CC tissues and ANCTs	Up-regulation of circ_0011385 (which sponges miR-149-5p)	_	higher FIGO stage	
Melanoma	_ melanoma tissue and ANCTs	Up-regulation of LRIG2 (a target of miR-149-5p)	_	_	[46]
Acute myeloid leukemia (AML)	45 AML, T cell ALL, and CML patients and 20 healthy controls	Up-regulation of miR-149-5p	_	_	[47]
Acute lymphocytic leukemia (ALL)	GSE166579	Dow-regulation of circADD2 (which sponges miR-149-5p)	_	_	[48]
	30 ALL patients and 30 controls	Dow-regulation of circADD2 (which sponges miR-149-5p)	_	_	
Pancreatic ductal adenocarcinoma (PDAC)	27 PDAC patients and 3 healthy controls	Up-regulation of miR-149-5p	_	_	[53]

## 5. Discussion

miR-149-5p has been shown to be sponged by a number of circRNAs and lncRNAs such as circNRIP1, circNHSL1, circCTNNA1, circ5615, circ-FLNA, circPVT1, circ_0072995, circ_0000654, circ_0075341, circRNA_100146, circ_0011385, circADD2, BLACAT1, LINC00460, PCAT1, DLEU1, DLGAP1-AS1, HOTAIR, MIAT, HNF1A-AS1, NEAT1 and SNHG8. Thus, the most appreciated mechanism of miR-149-5p dysregulation in cancer cells is the sequestering effects of these lncRNAs and circRNAs on it. Identification of the complex network between lncRNAs/circRNAs and miR-149-5p is a prerequisite for the development of targeted therapies for modulation of expression of this miRNA.

TGF-β2, Wnt/β-catenin, Hippo, TWEAK/EGFR and IL-6/STAT3 pathways are the main signaling pathways being regulated by miR-149-5p. Based on the results of Target Scan online tool (www.targetscan.org/cgi-bin/targetscan/vert_71/targetscan.cgi?species=Human&gid=&mir_c=miR-149-5p&mir_sc=&mir_nc=&sortType=cs&allTxs=&incl_nc=100, accessed on 28 November 2021), the top predicted targets of miR-149-5p are summarized in Table 4. Thus, miR-149-5p can regulate several independent cellular processes linked with immune function, gene expression and cell motility.

Then, we assessed the Kyoto Encyclopedia of Genes and Genomes (KEGG) pathways of miR-149-5p and its associated genes (Table 5).

Moreover, we retrieved genes and gene products associated with GO terms of miR-149-5p (Table 6).

Finally, we depicted an interaction network between miR-149-5p and lncRNAs (Figure 3).

The impact of miR-149-5p in carcinogenesis has been appraised by a number of in vitro and in vivo studies which have silenced this miRNA or the related lncRNAs/circRNAs. This miRNA can affect the responses of cancer cells to oxaliplatin, cisplatin, 5-fluouracil, sorafenib, gefitinib and trastuzumab.

Clinical studies have suggested a tumor suppressor role for mR-149-5p in gastric, liver, colorectal, medullary/papillary thyroid, breast, prostate, esophageal, renal, cervical and oral squamous cell cancers as well as osteosarcoma. In bladder and ovarian cancers, both tumor-suppressor and oncogenic effects have been reported for mR-149-5p. In lung cancer, while most studies have indicated a tumor-suppressor role for miR-149-5p, a single study has shown up-regulation of this miRNA in cancer patients compared with controls. Thus, most conducted studies are in favor of a tumor-suppressor role for mR-149-5p. However, a context-dependent role might be considered for this miRNA. Consistent with the tumor-suppressor role for this miRNA, several studies have shown correlation between the down-regulation of this miRNA and shorter survival of patients, indicating a role for miR-149-5p as a prognostic predictor.

Cumulatively, miR-149-5p partakes in a complex functional network which is constructed by several cancer-related lncRNAs and circRNAs. This network efficiently controls the activity of several cancer-related signaling pathways. Modulation of expression of miR-149-5p can be regarded as a therapeutic modality for the attenuation of cancer cells’ growth and the induction of chemosensitivity in these cells.

In conclusion, miR-149-5p is an example of miRNAs with important physiological roles whose expression has been dysregulated in various types of cancer. Future studies should focus on the design of targeted therapies for the amendment of dysregulation of this miRNA.

## Figures and Tables

**Figure 1 ijms-23-00415-f001:**
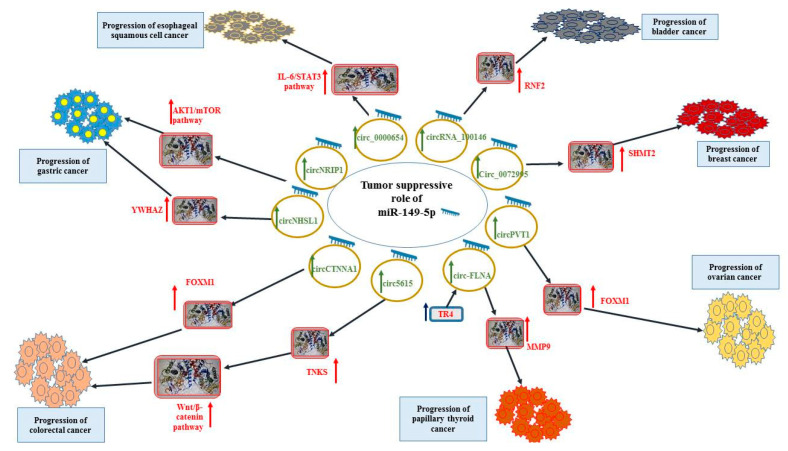
Tumor-suppressive role of miR-149-5p in esophageal cancer, bladder cancer, breast cancer, ovarian cancer, thyroid cancer and colorectal cancer.

**Figure 2 ijms-23-00415-f002:**
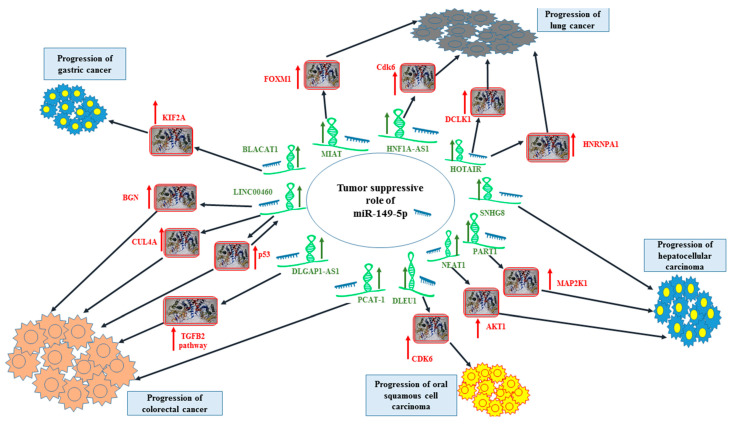
Tumor-suppressive role of miR-149-5p in gastric cancer, lung cancer, colorectal cancer, hepatocellular carcinoma and oral squamous cell carcinoma.

**Figure 3 ijms-23-00415-f003:**
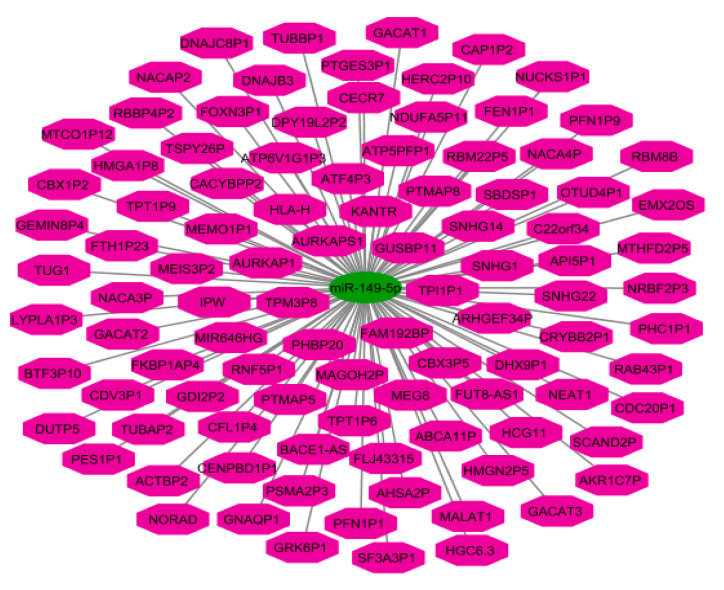
An interaction network between miR-149-5p and long non-coding RNAs (lncRNAs). The pink octagon and green ellipse indicate lncRNAs and miR-149-5p, respectively. The graph visualization was constructed by Cytoscape version 3.6.1.

**Table 1 ijms-23-00415-t001:** Role of miR-149-5p or miR-149-5p-interacting genes in carcinogenesis based on cell line studies (∆: knock-down or deletion, ↑: up-regulation or enhancement, ↓: down-regulation, DDP and CDDP: cisplatin, UA: ursolic acid, PTX: Paclitaxel).

Tumor Type	Targets/Regulators and Signaling Pathways that Interact with miR-149-5p	Function	Reference
Gastric cancer	circNRIP1, AKT1/mTOR pathway	∆ circNRIP1: 44001proliferation, ↓ invasion, ↓ migration	[6]
	BLACAT1, KIF2A	↑ miR-149-5p: ↓ proliferation, ↓ invasion, ↓ migration	[7]
	circNHSL1, YWHAZ	∆ circNRIP1: ↓ invasion, ↓ migration, ↓ glutaminolysis	[8]
Colorectal cancer	LINC00460, p53, miR-150-5p	∆ p53 or LINC00460: ↓ oxaliplatin resistance	[9]
	LINC00460, BGN	∆ LINC00460: ↓ invasion, ↓ migration	[10]
	circCTNNA1, FOXM1	↑ circCTNNA1: ↑ proliferation, ↑ invasion, ↑ migration, ↑ viability, ↑ colony-forming, ↑ DNA synthesis ∆ circCTNNA1: ↓ proliferation, ↓ viability, ↓ colony-forming, ↑ G1 arrest	[11]
	PCAT-1	∆ PCAT-1: ↓ proliferation, ↓ invasion, ↓ migration, ↑ apoptosis	[12]
	DLGAP1-AS1, TGFB2 signaling pathway	∆ DLGAP1-AS1: ↓ proliferation, ↓ invasion, ↓ migration, ↑ apoptosis, ↑ sensitivity to 5-FU ∆ miR-149-5p: ↑ proliferation, ↓ apoptosis	[13]
	circ5615, TNKS, Wnt/β-catenin pathway	∆ DLGAP1-AS1: ↓ proliferation, ↓ invasion ↑ DLGAP1-AS1: ↑ proliferation, ↑ transition from G1/S to G2/M phase	[14]
	LINC00460, CUL4A	∆ DLGAP1-AS1: ↓ proliferation, ↑ apoptosis, ↑ G0/G1 phase arrest	[40]
Medullary thyroid carcinoma	GIT1	∆ miR-149-5p: ↑ proliferation, ↑ invasion ↑ miR-149-5p: ↓ proliferation, ↓ invasion	[15]
Papillary thyroid cancer	circ-FLNA, TR4, MMP9	∆ TR4: ↓ proliferation, ↓ invasion, ↓ migration	[16]
Oral squamous cell carcinoma	TGFβ2	↑ miR-149-5p: ↓ proliferation, ↓ invasion, ↓ migration, ↓ CDDP resistance, ↑ apoptosis	[17]
	DLEU1, CDK6	∆ DLEU1: ↓ proliferation, ↑ G1 cell cycle arrest	[18]
Ovarian cancer	Hippo signaling pathway, MST1, SAV1	↑ miR-149-5p: ↓ apoptosis, ↑ resistance to CDDP ∆ miR-149-5p: ↑ apoptosis, ↓ resistance to CDDP	[19]
	circPVT1, FOXM1	∆ circPVT1: ↓ viability, ↓ invasion, ↓ migration	[20]
Osteosarcoma	TNFRSF12A, TWEAK/EGFR pathway	∆ miR-149-5p: ↑ proliferation, ↑ colony formation	[21]
Breast cancer	Circ_0072995, SHMT2	∆ Circ_0072995: ↓ proliferation, ↓ invasion, ↓ migration, ↓ anaerobic glycolysis↑ apoptosis ∆ SHMT2: ↓ proliferation, ↓ invasion, ↓ migration, ↓ anaerobic glycolysis↑ apoptosis	[22]
	IL-6	∆ miR-149-5p: ↓ propofol effects (↓ sensitization to trastuzumab)	[23]
	MyD88	UA treatment: ↑ miR-149-5p = ↓ PTX resistance	[24]
Lung cancer	B3GNT3	∆ B3GNT3: ↓ proliferation, ↓ invasion, ↓ colony formation ↑ miR-149-5p: ↓ proliferation, ↓ tumorigenesis	[27]
	HOTAIR, HNRNPA1	↑ miR-149-5p: ↓ proliferation, ↓ invasion, ↓ migration, ↑ G0/G1 phase arrest, did not affect apoptosis	[28]
	HNF1A-AS1, Cdk6	∆ HNF1A-AS1: ↓ proliferation, ↓ invasion, ↓ migration	[29]
	MIAT, FOXM1	∆ MIAT: ↓ proliferation, ↓ invasion, ↓ migration, ↓ colony formation, ↑ G1 phase arrest	[30]
	_	∆ miR-149-5p: ↓ gefitinib resistance, ↑ apoptosis ↑ miR-149-5p: ↑ proliferation, ↑ gefitinib resistance	[33]
	HOTAIR, DCLK1	∆ HOTAIR: ↓ proliferation, ↓ invasion, ↓ migration, ↓ DDP resistance	[31]
	AMTOL2	↑ miR-149-5p: ↑ proliferation, ↓ apoptosis ∆ miR-149-5p: ↓ proliferation, ↑ apoptosis	[34]
	LINC00460, IL-6	∆ LINC00460: ↓ EMT process, ↓ migration, ↓ viability following treatment with various EGFR-TKIs, ↑ apoptosis following treatment with various EGFR-TKIs	[32]
Hepatocellular carcinoma	MMP9	co-culture: PMA + IL-4 and IL-13: ↑ invasion, ↑ migration	[35]
	NEAT1, AKT1	∆ NEAT1: ↑ sensitivity to sorafenib, ↑ apoptosis, ↓ viability	[36]
	PART1, MAP2K1	∆ PART1: ↓ proliferation, ↓ invasion, ↓ migration ↑ miR-149-5p: ↓ proliferation, ↓ invasion, ↓ migration	[37]
	SNHG8	∆ SNHG8: ↓ proliferation, ↓ invasion, ↓ migration, ↓ viability, ↓ colony formation	[41]
	MTHFR, TP53INP1, PDCD4	FA deficiency: ↑ miR-149-5p	[38]
Prostate carcinoma	RGS17	↑ miR-149-5p: ↓ proliferation, ↓ migration, ↓ viability	[25]
Esophageal squamous cell cancer	circ_0000654, IL-6/STAT3 signaling pathway	∆ circ_0000654: ↓ proliferation, ↓ invasion, ↓ migration, ↑ apoptosis ↑ circ_0000654: ↑ proliferation, ↑ invasion, ↑ migration, ↓ apoptosis ↑ miR-149-5p: ↓ proliferation, ↓ invasion, ↓ migration, ↑ apoptosis	[39]
Bladder cancer	circRNA_100146, RNF2	∆ circRNA_100146: ↓ proliferation, ↓ invasion, ↓ migration, ↑ apoptosis	[26]
Renal cell carcinoma	_	↑ miR-149-5p: ↓ proliferation, ↓ migration, ↑ apoptosis	[42]
	FOXM1	↑ miR-149-5p: ↓ proliferation, ↓ invasion, ↓ migration	[43]
Cervical cancer	circ_0075341, AURKA	∆ circ_0075341: ↓ proliferation, ↓ invasion	[44]
	circ_0011385, SOX4	∆ circ_0011385: ↓ multiplication, ↓ invasion, ↓ migration	[45]
Melanoma	LRIG2	↑ miR-149-5p: ↓ colony formation, ↑ apoptosis	[46]
Acute myeloid leukemia	FASLG, p-FADD, caspases	∆ miR-149-5p: ↑ apoptosis	[47]
Acute lymphocytic leukemia	circADD2, AKT2	↑ circADD2: ↓ proliferation, ↓ miR-149-5p	[48]

**Table 2 ijms-23-00415-t002:** Role of miR-149-5p or miR-149-5p-intercating genes in the carcinogenesis based on animal models (∆: knock-down or deletion, ↑: up-regulation or enhancement, ↓: down-regulation, CDDP: cisplatin, UA: ursolic acid, PTX: Paclitaxel).

Tumor Type	Animal Models	Results	Reference
Gastric cancer	BALB/c nude mice	∆ circNRIP1 (which sponges miR-149-5p): ↓ tumor volume, ↓ tumor weight	[6]
	female BALB/c nude mice	∆ BLACAT1 (which sponges miR-149-5p): ↓ tumor volume, ↓ tumor weight	[7]
	male BALB/C nude mice	∆ circNHSL1 (which sponges miR-149-5p): ↓ tumor volume, ↓ tumor weight	[8]
Colorectal cancer	BALB/c nude mice	∆ LINC00460 (which sponges miR-149-5p): ↓ tumor volume, ↓ tumor weight	[9]
	female BALB/c nude mice	↑ circCTNNA1 (which sponges miR-149-5p): ↑tumor volume, ↑tumor weight	[11]
	female nude mice	∆ DLGAP1-AS1 (which sponges miR-149-5p): ↓tumor growth, ↑5-FU sensitivity	[13]
	male BALB/C nude mice	∆ circ5615 (which sponges miR-149-5p): ↓ tumor growth	[14]
	athymic male mice	∆ LINC00460 (which sponges miR-149-5p): ↓tumor volume, ↓tumor weight	[40]
Papillary thyroid cancer	male nude mice	∆ circ-FLNA (which sponges miR-149-5p): ↓tumor growth, ↓tumor volume, ↓ metastasis	[16]
Ovarian cancer	BALB/c-nu mice	∆ miR-149-5p: ↓ tumor volume, ↓ tumor weight, ↓ resistance to CDDP	[19]
Breast cancer	female BALB/c nude mice	∆ circ_0072995: ↓ tumor volume, ↓ tumor weight	[22]
	female athymic mice	∆ miR-149-5p: ↓ ability of propofol to inhibit lung metastasis	[23]
	female athymic nude mice	UA and PTX treatment: ↓tumor volume, ↓ tumor weight	[24]
Lung cancer	BALB/c nude mice	∆ B3GNT3: ↓ tumor volume, ↓ tumor weight	[27]
	female BALB/c nude mice	∆ HNF1A- AS1: ↓ tumor volume, ↓ tumor weight	[29]
	female BALB/c nude mice	∆ MIAT: ↓ tumor volume, ↓ tumor weight	[30]
	athymic BALB/c mice	∆ HOTAIR: ↓ tumor volume, ↓ tumor weight, ↓ DDP resistance	[31]
Hepatocellular carcinoma	nude mice	↑ macrophages (THP-1): ↑ metastasis miR-149-5p: ↓ metastasis	[35]
	nude mice	∆ SNHG8: ↓ tumor volume, ↓ tumor weight, ↓ metastasis	[41]
Esophageal squamous cell cancer	male BALB/c athymic nude mice	∆ circ_0000654: ↓ tumor volume, ↓ tumor weight, ↓ metastasis	[39]
Acute lymphocytic leukemia	BALB/c nude mice	↑ circADD2: ↓ tumor volume, ↓ tumor weight	[48]

**Table 4 ijms-23-00415-t004:** Top predicted targets of miR-149-5p.

Rget Gene	Representative Transcript	Gene Name	Number of 3P-seq Tags Supporting UTR + 5	Conserved Sites				Poorly Conserved Sites				6mer Sites	Cumulative Weighted Context++ Score	Total Context++ Score
				Total	8mer	7mer-m8	7mer-A1	Total	8mer	7mer-m8	7mer-A1			
*LRIG2*	ENST00000361127.5	leucine-rich repeats and immunoglobulin-like domains 2	348	1	1	0	0	3	1	2	0	1	−0.96	−1.08
*STRADB*	ENST00000392249.2	STE20-related kinase adaptor beta	12	1	0	1	0	1	1	0	0	1	−0.87	−0.87
*ZBTB37*	ENST00000367701.5	zinc finger and BTB domain containing 37	213	0	0	0	0	5	0	1	4	4	−0.83	−0.92
*TTLL9*	ENST00000375921.2	tubulin tyrosine ligase-like family, member 9	5	0	0	0	0	3	2	0	1	1	−0.82	−0.82
*FRMD7*	ENST00000370879.1	FERM domain containing 7	5	0	0	0	0	2	1	1	0	0	−0.81	−0.81
*TSPAN31*	ENST00000547992.1	tetraspanin 31	72	0	0	0	0	2	1	1	0	2	−0.78	−1.10
*CTRC*	ENST00000375943.2	chymotrypsin C (caldecrin)	5	0	0	0	0	3	1	1	1	2	−0.77	−0.77
*STRA6*	ENST00000395105.4	stimulated by retinoic acid 6	42	0	0	0	0	4	1	3	0	1	−0.77	−0.77
*GUCD1*	ENST00000447813.2	guanylyl cyclase domain containing 1	2990	0	0	0	0	2	2	0	0	1	−0.77	−0.77
*KCNJ5*	ENST00000529694.1	potassium inwardly-rectifying channel, subfamily J, member 5	12	0	0	0	0	4	0	4	0	2	−0.75	

**Table 5 ijms-23-00415-t005:** Kyoto Encyclopedia of Genes and Genomes (KEGG) pathways analyses of miR-149-5p and its associated genes were extracted from the EVmiRNA: the extracellular vesicles miRNA database.

KEGG ID	KEGG Description	Genes	*p* Value
ko04962	Vasopressin-regulated water reabsorption	*VAMP2, RAB5C, ADCY9, DCTN2, DCTN4, DCTN5*	2.45 × 10^−52^
ko04380	Osteoclast differentiation	*ITGB3, SOCS1, SOCS3, FOS, IFNG, JUN, GAB2, CSF1, IFNAR2, SYK, GRB2, IFNGR2, MITF, IL1A, IL1B, TGFB2*	1.47 × 10^−118^
ko05218	Melanoma	*PDGFRB, PDGFRA, TP53, RB1, CDK6, MITF, FGFR1*	4.09× 10^−58^
ko05210	Colorectal cancer	*TCF7, TP53, AXIN1, FOS, BIRC5, JUN, TGFB2, MSH6, LEF1*	1.58× 10^−70^
ko05332	Graft-versus-host disease	*IFNG, IL6, IL1B, PRF1, IL1A*	3.12× 10^−28^
ko05202	Transcriptional misregulation in cancers	*MEN1, TAF15, NCOR1, FLT1, CEBPA, SPINT1, ZBTB16, SIX4, GOLPH3, AFF1, KLF3, JMJD1C, TRAF1, SP1, PLAU, PBX1, TP53, CCND2, IL6, ELK4, MMP9, BCL6, EWSR1*	4.43× 10^−160^
ko05142	Chagas disease (American trypanosomiasis)	*TGFB2, IRAK1, ADCY1, FOS, IFNG, CALR, JUN, NOS2, IL6, IFNGR2, IL1B, MYD88, CD3E, IRAK4, GNAQ*	3.41× 10^−103^
ko05140	Leishmaniasis	*MARCKSL1, ITGB1, TGFB2, MYD88, FOS, IFNG, JUN, NOS2, IFNGR2, IL1A, IL1B, IRAK1, IRAK4*	8.32× 10^−71^
ko05412	Arrhythmogenic right ventricular cardiomyopathy (ARVC)	*ITGB1, ITGA9, ITGB3, SGCD, GJA1, CACNB2, CACNA1C, CACNA1D, TCF7, LEF1, CACNG4*	5.27× 10^−89^
ko04540	Gap junction	*GJA1, PDGFRA, GNAQ, PDGFRB, DRD2, ADCY9, SRC, GRB2, ADCY1*	7.36× 10^−78^
ko03015	mRNA surveillance pathway	*SMG5, PABPN1, SMG6, RNPS1, SYMPK, CSTF2, SAP18, UPF2, ACIN1, RNMT*	8.17× 10^−83^
ko04930	Type II diabetes mellitus	*SOCS4, CACNA1C, SOCS1, SOCS3, SLC2A4, CACNA1D*	1.12× 10^−50^

**Table 6 ijms-23-00415-t006:** Genes and gene products associated with GO terms of miR-149-5p extracted from AmiGO 2. ARUK-UCL: The Alzheimer’s Research UK University College London, BHF-UCL: British Heart Foundation UK University College London.

GO Class	Contributor	Reference
negative regulation of tumor necrosis factor production	ARUK-UCL	PMID:23595570
negative regulation of cell migration involved in sprouting angiogenesis	BHF-UCL	PMID:24463821
negative regulation of blood vessel endothelial cell proliferation involved in sprouting angiogenesis	BHF-UCL	PMID:24463821
negative regulation of endothelial cell chemotaxis to fibroblast growth factor	BHF-UCL	PMID:24463821
mRNA binding involved in posttranscriptional gene silencing	ARUK-UCL	PMID:23595570
negative regulation of epithelial to mesenchymal transition	ARUK-UCL	PMID:25916550
negative regulation of cell migration	ARUK-UCL	PMID:26498692
negative regulation of interleukin-6 production	ARUK-UCL	PMID:24299952
positive regulation of transforming growth factor beta3 production	ARUK-UCL	GO_REF:0000024
positive regulation of collagen biosynthetic process	ARUK-UCL	GO_REF:0000024
gene silencing by miRNA	ARUK-UCL	PMID:23595570
negative regulation of fibroblast growth factor receptor signaling pathway	BHF-UCL	PMID:24463821
cellular response to fibroblast growth factor stimulus	BHF-UCL	PMID:24463821
negative regulation of inflammatory response	ARUK-UCL	GO_REF:0000024
negative regulation of stress fiber assembly	ARUK-UCL	PMID:26498692
negative regulation of NIK/NF-kappaB signaling	ARUK-UCL	GO_REF:0000024

## Data Availability

Not applicable.

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
