# Peer review of "A Review on the Role of miR-149-5p in the Carcinogenesis"

_ijms, 2021, doi:10.3390/ijms23010415_

Round 1

Reviewer 1 Report

In the review paper authors summarized role of miRNA-149 in carcinogenesis. Paper is a little bit chaotic and cursory and should be reconstructed.

  1. There are too many figures which are unnecessary. It will be better to compose one of two more informative ones, that for instance merge meachnisms similar ofr the different cancers.
  2. To understand better role of the miRNA, some analysis withe the use of online bioinformatics tools should be conducted, including the common, such as GO and KEGG.
  3. Tablke 1 and 3 are very spacious, the better idea will be to reduce them to more readable points (too many data)
  4. There are very scant data concerning the clinical findings, almost all of data is summarized in table. Some clinical evidences, such as AUC, OS, PFS or HRs should be provided.

Author Response

  1. We designed tow new figures merging the previous figures.
  2. We added results of an in silico target prediction.
  3. We edited these tables and deleted a complete column from each table.
  4. We added some data about OS and PFS.

Reviewer 2 Report

The manuscript by Ghafouri-Fard et al. presents the role of miR-149-5p in carcinogenesis. Although the article is of medium interest for the specialists in the field in presents an in depth evaluation of this miRNA. There are some issues that need to be addressed before moving forward. 

- the manuscript is hard to follow because of the long sentences and unclear message of some phrases ( ex: line 28-30)

  • keywords : miR-149-5p not miR-149-59
  • the introduction is practically the same as the abstract - it doesn't bring any in depth information about miR-149-5p, only very general aspects
  • the article needs to have a better organization meaning that the implication of miR-149-5p should appear in gastrointestinal tumors; endocrine tumors, etc. in each chapter for the readers to have a easy experience while reading
  • the discussion section needs to be rewritten to point out and compare data ( at the moment is just a simple summary)
  • discuss about future perspectives and potential of this miRNA
  • the text needs to be revised for English
  • state a conclusion of your findings
  • discuss also about miR-149-3p as comparing the data from the literature

For a review article is not enough to comprise the data in a series of tables and figures. The work needs more thinking and organisation. 

Author Response

  1. We edited the mentioned sentence (lines 28-30).
  2. We corrected the mentioned point in key words.
  3. We edited the Introduction.
  4. We added subtitles.
  5. We edited the discussion.
  6. We edited the language of the manuscript.
  7. We added conclusion.

Round 2

Reviewer 1 Report

However authors made effort to improve the paper quality, it has not reach appropriate priority score. Some points suggested by reviewer were considered,, but not all.

Point 2. Target prediction was not prepared according to suggestions, there is lack of interaction between miRNA and other RNAs, pathway and molecular, cellular processes.
Point 3. Tables are still unclear and data presented in confusing way that is difficult to interpretation

Author Response

We added tables and a figure showing interaction between miRNA and other RNAs, pathway and molecular, cellular processes.
We edited Tables.

Reviewer 2 Report

After the current modification the article is suitable for publication. 

Author Response

We edited the language of the manuscript.

Thank you for your approval.

Round 3

Reviewer 1 Report

Authors corrected paper according to all suggestions.